# Landslide Mapping and Susceptibility Assessment Using Geospatial Analysis and Earth Observation Data

**Emmanouil Psomiadis [1],\*, Andreas Papazachariou [1], Konstantinos X. Soulis [1], Despoina-Simoni Alexiou [1] and Ioannis Charalampopoulos [2]**

[1] Department of Natural Resources Management and Agricultural Engineering, Agricultural University of Athens, 75 Iera Odos st., 11855 Athens, Greece; papazacharioua@gmail.com (A.P.); soco@aua.gr (K.X.S.); sim.alexiou@aua.gr (D.-S.A.)

[2] Department of Crop Science, Agricultural University of Athens, 75 Iera Odos st., 11855 Athens, Greece; icharalamp@aua.gr

\* Correspondence: mpsomiadis@aua.gr; Tel.: +30-210-529-4156

**Abstract:** The western part of Crete Island has undergone serious landslide events in the past. The intense rainfalls that took place in the September 2018 to February 2019 period provoked extensive landslide events at the northern part of Chania prefecture, along the motorway A90. Geospatial analysis methods and earth observation data were utilized to investigate the impact of the various physical and anthropogenic factors on landslides and to evaluate landslide susceptibility. The landslide inventory map was created based on literature, aerial photo analysis, satellite images, and field surveys. A very high-resolution Digital Elevation Model (DEM) and land cover map was produced from a dense point cloud and Earth Observation data (Landsat 8), accordingly. Sentinel-2 data were used for the detection of the recent landslide events and offered suitable information for two of them. Eight triggering factors were selected and manipulated in a GIS-based environment. A semi-quantitative method of Analytical Hierarchy Process (AHP) and Weighted Linear Combination (WLC) was applied to evaluate the landslide susceptibility index (LSI) both for Chania prefecture and the motorway A90 in Chania. The validation of the two LSI maps provided accurate results and, in addition, several susceptible points with high landslide hazards along the motorway A90 were detected.

**Keywords:** landslide susceptibility mapping; remote sensing and GIS-based analysis; rainfall and anthropogenic-triggered events; hazard assessment

## 1. Introduction

Landslides are regarded among the most hazardous and recurrently appearing natural disasters globally [1–5]. The triggering factors that cause landslide episodes are numerous and complexly interdependent, comprising intensive rainfalls, earthquakes, rapid stream erosion, geomorphological processes and human activities (deforestation of slopes, road construction, uncontrolled irrigation, quarries and mines, etc.) [6–8]. Yet foreseeing the place and time that a landslide will take place constitutes a complex issue, since the properties of geology and slope angle characteristics vary greatly over short distances, and the timing, location, and intensity of triggering events are difficult to be estimated [6].

Rainfall-triggered landslides are amongst the most frequent types, as well as the most devastating hazards, which cause extensive devastation to lives, properties and infrastructures worldwide [9–13]. The intensity and duration of rainfall can influence slope balance by putting extra weight into the soil

and creating considerable pore pressure [14]. In general, the crucial precipitation that affects slope stability is determined by the meteorological conditions and geomorphological characteristics [14,15]. Moreover, the landslides that follow human activity, and are specifically the result of infrastructure and road construction, have expanded over time, due to the deficiency of the sufficient rating of slopes, an adequate configuration of drainage features and consideration of past occasions [5,15].

The critical necessity to better mitigate and prevent geohazards led to the development of Landslide Susceptibility (LS) assessment methods [13,16–20]. An LS map represents assured places with the propensity of future landslide phenomena by taking into account some of the most important parameters that are related to landslides and utilizing the previous spatial distribution of landslide occasions as inputs [17,21–24]. A precise susceptibility recording could be the decisive material for a wide range of operators and managers from governmental departments and the scientific community to achieve hazard reduction [24,25]. LS analysis principally includes four distinct stages: (i) the creation of a landslide inventory map; (ii) the estimation of preparatory and triggering factors that affect the landslide occurrence; (iii) the selection of suitable methodologies for designating the factors' weights and (iv) the calculation of the LS map [26].

There are three LS assessment approaches, the qualitative, the quantitative and the semi-quantitative [7,17,27–34]. Qualitative approaches are, in fact, very subjective insofar as they depend on knowledge of the experts. Moreover, they utilize the landslide index to classify regions with similar geomorphological and lithological features, indicating high susceptibility to landslides [33,35,36]. Quantitative methods produce numerical evaluations among triggering parameters and landslides and are classified into two types: deterministic and statistical [23]. As far as it concerns these procedures, historical landslides can be correlated to quantifiable characteristics of the landscape and can be used to estimate the potential forthcoming events [17,27,36–40]. Nevertheless, several qualitative methods that make use of weighting processes are generally characterized as semi-quantitative techniques [38,41–43], such as the analytic hierarchy process (AHP) [2,38,39,44–46] and the weighted linear combination (WLC) [43,47]. Several LS mapping methodologies have been introduced over the last few decades [17,38]. They aim to categorize several parts of the landscape following the grade of landslide susceptibility [2]. In recent years, Earth Observation (EO) platforms and Geographical Information System (GIS) tools have been efficiently utilized for hazard detection, monitoring and susceptibility assessment [12,48]. EO datasets, such as Landsat-8 (NASA/USGS program) and Sentinel-2 (European Space Agency's Copernicus program) [49,50], and GIS have proven their capability for landslide research. More specifically, they provide information and tools to estimate destruction due to landslides [51], to prepare an inventory of landslides [52,53], to assess the landslide volume by decoding the activating parameter and mechanism of collapse [54,55] and to monitor landslide hazards for prediction and mitigation objectives [26,38,56–58]. Digital Elevation Model (DEM) generation technology from point clouds derived from very high resolution (VHR) aerial imagery allows us to produce a wide range of detailed topographic data promptly, improving the efficiency of data acquisition significantly [59,60].

Furthermore, landslide hazard (LH) estimation constitutes a significant action in the devastation risk controlling approximation. Hazard evaluation necessitates the designation of the degree or intensity of an episode over time [61–63]. The primary and most vital stage for LH assessment is signified by the correlation of the landslide inventory geodatabase (GB). A well-defined GB gives extensive information of the place and nature of previous landslide events, the failure mechanisms, the triggering parameters, the event incidence, the dimensions, and the destruction that occurred [18,63,64].

The area under research is Chania Prefecture, which lies in the western part of Crete Island [65,66]. Chania prefecture is characterized by tectonic deformation and high seismic activity (due to the adjacent Hellenic fore-arc). It has undergone severe landslide phenomena both in the past and in recent years. The establishment of an LS map in such a vulnerable area could make the risk administration attainable for the researchers and civil protection authorities.

The study aims to identify the impact of the various physical (mainly rainfall-triggered phenomena) and anthropogenic factors (mainly road constructions and their characteristics) on LS at the Chania prefecture, Crete, an area that has been highly vulnerable to landslide phenomena in the distant and recent past. For this purpose, the Analytic Hierarchy Process (AHP) for Weighted Linear Combination (WLC) approach was utilized. Through this research, we intend (i) to examine the suitability of the EO and derived DEM data in landslide detection and LS assessment, (ii) to evaluate the relation of Chania prefecture landslide susceptibility with the historical landslide occurrences and the recent rainfall-triggered landslide events (during 2018-2019 period), and (iii) to assess the potential landslide hazard assessment of analogous regions and (iv) to make suggestions for the prevention and mitigation of new catastrophic landslides in the future.

## 2. Study Area

The prefecture of Chania situated at the western part of Crete Island, located between latitudes 35°11′14.2′′ and 36°11′10.6′′ N, and longitudes 23°15′19.6 and 24°30′20.1 E, in South Aegean Sea and covers an area of almost 2,376 km$^2$ and population of 156,585 residents (Figure 1) [67]. The study area was limited to approximately 98% of the entire prefecture of Chania due to lack of data in the southeastern part of the prefecture [68–70].

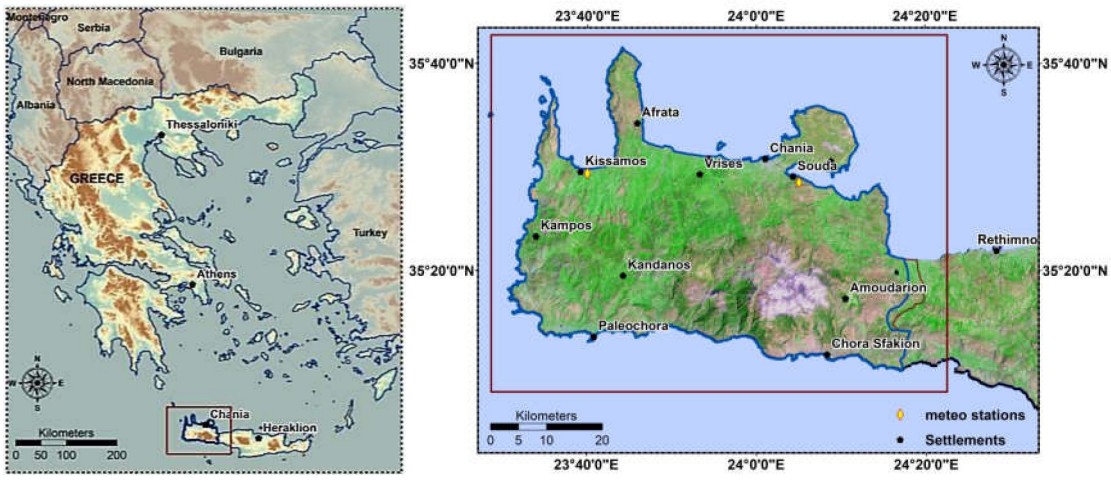

**Figure 1.** Location of the study area (red rectangle).

Crete island is characterized by active tectonics which generate abrupt geomorphology and lithological compound formations. It consists of pre-alpine, alpine and post-alpine formations and is characterized by the existence of complex tectonic and geological characteristics. In western Crete, the geological stratigraphic sequence consists of the following tectonic units: (i) Pindos unit, (ii) Tripoli unit, (iii) Phyllite–Quartzite nappe (iv) Trypali carbonate unit and (v) Plattenkalk limestone. The Neogene and Quaternary sediments are placed on top of the complete sequence. The Quaternary sediments contain loose clay sands, gravels, and sandstones of various compositions. This formation outcrops along the coastline, in streams, and within small internal basins. The Neogene comprises interbedded layers of marl and marly limestone of a mean thickness of some 1–2 m centimetres. Sandstone, conglomerate, and sandy clays can be found within the formation. Flysch is present in the western and southwestern parts of the island. The limestone of the Pindos tectonic unit appears to a small extent in the west Chania region overlying on top of the Tripoli unit. The Tripoli tectonic unit in the Chania area is over-thrusted on top of the Phylites' unit, Trypali unit, or over the Plattenkalk limestone (in SE region). It comprises mainly limestone, and the lower part of the sequence is fragmented as a result of the tectonic environment. The limestone is karstified, which results in steep relief and a weak drainage system, most of the time driven by the dominant tectonic structures. Under the

Tripoli unit, Phyllites–Quartzites seem to be extensively deformed and are characterized by isoclinal folds with axes trending NE-SW and NW-SE, dipping towards N or NE, and SE, implying a high pressure/low-temperature metamorphic phase. It consists of quartz-rich formation with interbedded layers of limestone, gypsum, and volcanic rocks [71], while the Trypali tectonic nappe comprises carbonates in the central part of the study area. The tectonic environment represented by faults can be grouped in two main classes: (i) faults trending NW-SE and (ii) faults trending W-E [72].

The geomorphology of Chania prefecture has the characteristics of the relief of the whole of Crete, which is crossed from west to east by a continuous mountain range, which is interrupted by valleys and canyons. The mountainous area (Mt Lefka Ori, 2,453m) covers almost the entire central and southern part of the region, while the plain (18.3% of the total prefecture) occupies the north coastal area. The main torrents of the area are located in the northern part, from which the most important are the rivers Kiliaris, Kladissos, Tavronitis, Tiflos and Keritis. The climate of the prefecture is characterized as temperate Mediterranean, with mild winters and hot and dry summers. Temperature fluctuations are highest in the Lefka Ori region, and snowfall is highest in November. Rainfall is higher than in the rest of Crete, due to the Lefka Ori and the orographic effect. The mean annual precipitation for the region has been assessed to vary from 650 to 1000 mm in the plains and mountainous areas, respectively (Figure 2) [73,74].

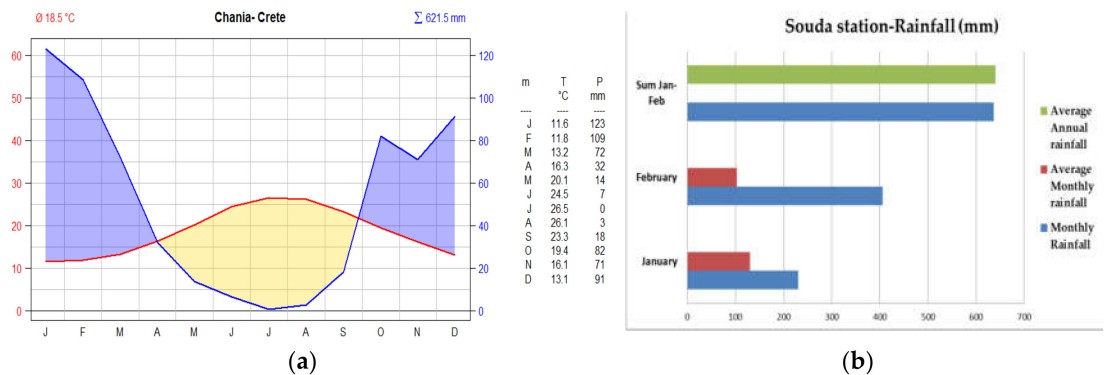

**Figure 2.** (**a**) Walter–Lieth diagram for Chania region illustrates the climatic variables of air temperature and precipitation, (**b**) The average annual, average monthly (January-February) and monthly rainfall amounts that recorded at Souda Weather Station [75].

*2.1. Intense Rainfall Period*

Western Crete, during the period from September 2018 until the end of February 2019, received significant rainfall amounts. It is noteworthy that the number of rainy days was very high in January, reaching up to 24 days in almost all areas of the island (typically ranges from 10 to 12 days), due to the continuous movement of barometric systems from the north and west. Notably, the recordings of rainfalls in the areas of Chania exceeded the rainfalls recorded in western and northern parts of Greece, where the highest rainfall levels are usually observed. The highest amounts of rain occurred at the mountainous areas (i.e., the area of Asi Gonia has the highest rainfall record with 1,963 mm of which 1,328 mm was recorded from September to December 2018 and 635 mm in January 2019).

During February 2019 two severe weather systems, "Hioni" and "Okeanis" took place, causing flooding and landslides, which brought about five fatalities and extensive floods that caused extensive damages to infrastructures and agricultural land. Between the 1 and 26 February, 269 mm of rain was recorded at the weather station of Kastelli (Figure 1), which constitutes the 270% of the regular average monthly rainfall of February, while the overall rainfall quantity for the two months (January and February 2019) was 497 mm, that is the 72% of the average yearly precipitation amounts. At Souda weather station (Figure 1), 116 mm of rainfall was recorded on 25 February, which exceeds the average monthly rainfall for the whole month, whilst in the period from 1 to 26 February 406mm was measured, which is four times the average monthly rate (Figure 2). In general, during February, some regions of

western Crete received amounts of water three or four times more than the average monthly rainfall. Particularly in the region of Askiftos, the total monthly rain and snow was above the European highest monthly rainfall height [75,76].

*2.2. Landslides*

In Greece, several landslide events have been recorded over the years (mainly situated at the western part) due to the existence of several preparatory parameters such as the complex geological formations (with intense folding and jointing) and the steep slopes. The triggering factors which are related to these phenomena are principally the intense rainfalls and earthquakes [14]. The national pattern of the landslide events (70% of the recorded numbers) in accordance with the form of landslides reveals that are primarily translational and rotational earth slides [14,22,24,77]. The rest of the failures comprise various types of debris flows and rockfalls. The Chania prefecture suffered from severe landslide phenomena in the past, primarily related to rotational and translational types (70–80%) and less so to the rockfall type, principally due to abrupt and mountainous geomorphology (Figure 3a) [14,24]. Intensive rainfall is commonly the most considerable triggering factor for their origination.

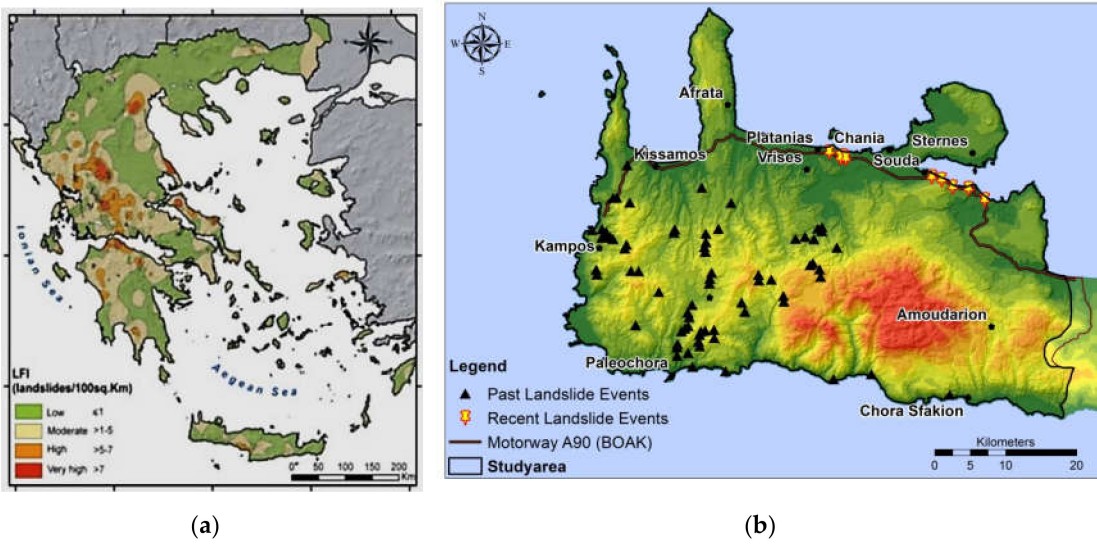

(**a**)                    (**b**)

**Figure 3.** (**a**) The Landslide Frequency Index (LFI-landslides/100 km$^2$) map of Greece [22], (**b**) The past and recent (2018–2019) landslide events in Chania prefecture [23].

The intense and prolonged precipitations from the September 2018 to February 2019 period expanded the landslide phenomenon to areas in the north and north-east and particularly in areas of intense human activity that rarely have faced it in the past. All of the cases took place along the main highway (A90), which crosses the entire northern part of the island, from one end to the other, with an E-W direction of the prefecture, which constitutes high risk areas with intense human exposure and high vulnerability level (Figure 2b). More specifically, five (5) landslide events occurred in the west, close to Agia Marina (two sites), and in Stalos, Galatas and Katos Stalos (Malindreto) villages, and four at the east, close to Platani (three sites) and the Kalami area of the City of Chania [72] (Figure 3b). Most of these recent landslide events that took place along the main highway A90 are caused, along with the intense rainfalls, from slope instability problems. Hasty road constructions (because of time and financial constraints) without proper geological and geotechnical examination may lead, under extreme rainfall conditions, to slope failures [78].

## 3. Materials and Methods

### 3.1. Data

In Table 1 presented the study's different types of datasets. The geological maps were used to depict the prefecture's lithological and structural units, the topographic maps to develop the drainage and road network, the satellite and orthophoto images to extract the land cover types and supplement a detailed road network where needed (highway and main provincial roads) and in addition to create an extremely accurate DEM (2 m spatial resolution), and the precipitation records to estimate the historical mean annual precipitation for the period of 1971–2000 [79]. Several DEM-derived spatial maps were created, such as slope angle and slope aspect. The analysis of the geospatial data was accomplished utilizing ArcGIS (10.7.1, Environmental Systems Research Institute, Redlands, CA, USA) software.

**Table 1.** Datasets.

| Data | Data Characteristic | Acquisition Date/ Date of Creation | Usage |
|---|---|---|---|
| Geological maps | Institute of Geology and Mineral Exploration (IGME) at 1:50,000 scale/map sheets: Chania, Kastellion, Perivolia (Platanias), Paleochora, Rethymno, Sellia, Vrisses, Alikianos | 1971, 1970, 1956, 2000, 1988, 1982, 1993, 1969 | Geological formations and faults |
| Topographic maps | Hellenic Military Geographical Service at 1:50.000 scale/map sheets: Chania, Kastellion (Kissamos), Perivolia, Paleochora, Rethymno, Sellia, Vrisses, Vatolakos | 1993–1994 | Drainage network |
| Meteorological data | Hellenic National Meteorological Service. | 1971–2000 | Rainfall distribution |
| Landsat 8 | Operational Land Imager (OLI)—11 spectral bands—geometrically and atmospherically corrected—15m spatial resolution, (path/row: 182/035 and 182/036) | 14/08/2019 | Land Cover/ Normalized Difference Vegetation Index |
| Sentinel-2 | MSI–13 spectral bands—geometrically and atmospherically corrected (2A)—10m spatial resolution | 06/01/2018<br>11/01/2019<br>17/03/2019 | Landslide detection |
| Orthophoto maps | Aerial photographs orthorectified—1m spatial resolution | 2014 | Drainage and road network integration / Point Cloud for DEM |

### 3.2. Methodology

The applied methodology was divided into a six-stage process: (a) construction of the inventory map, (b) process of earth observation data, (c) formation of the DEM from point cloud manipulation, (d) selecting the appropriate factors and building a geodatabase, (e) weighting the factors and calculating the landslide susceptibility map for (i) Chania prefecture and (ii) along the main highway A90 and (f) evaluate the areas with potential future hazard to a landslide event along the A90 motorway.

#### 3.2.1. Landslide Inventory Map

A detailed landslide inventory map first and foremost demonstrates the location, time, type and number of landslides and therefore is crucial for understanding the correlation concerning landslide occurrence and causative parameters and to create an integrated hazard map [17,39,63,80–82].

The landslide inventory map assortment of Chania prefecture was developed using the following steps: (a) collection of landslides historical information from previous studies and literature (Figure 3b);

(b) fieldwork for the mapping of landslides of the recent events (Figure 4) and (c) detection of landslides on satellite images (Sentinel-2 and VHR Google Earth (Figure 4). A number of landslide sites (80 in total, 71 old and 9 recent) were collected throughout the study area. The created landslide inventory map according to random sampling was separated into two datasets: 80% (64 landslide spots) for the triggering factors selection and 20% (16 landslide spots) of the landslides for the validation process, respectively [8,81,83,84].

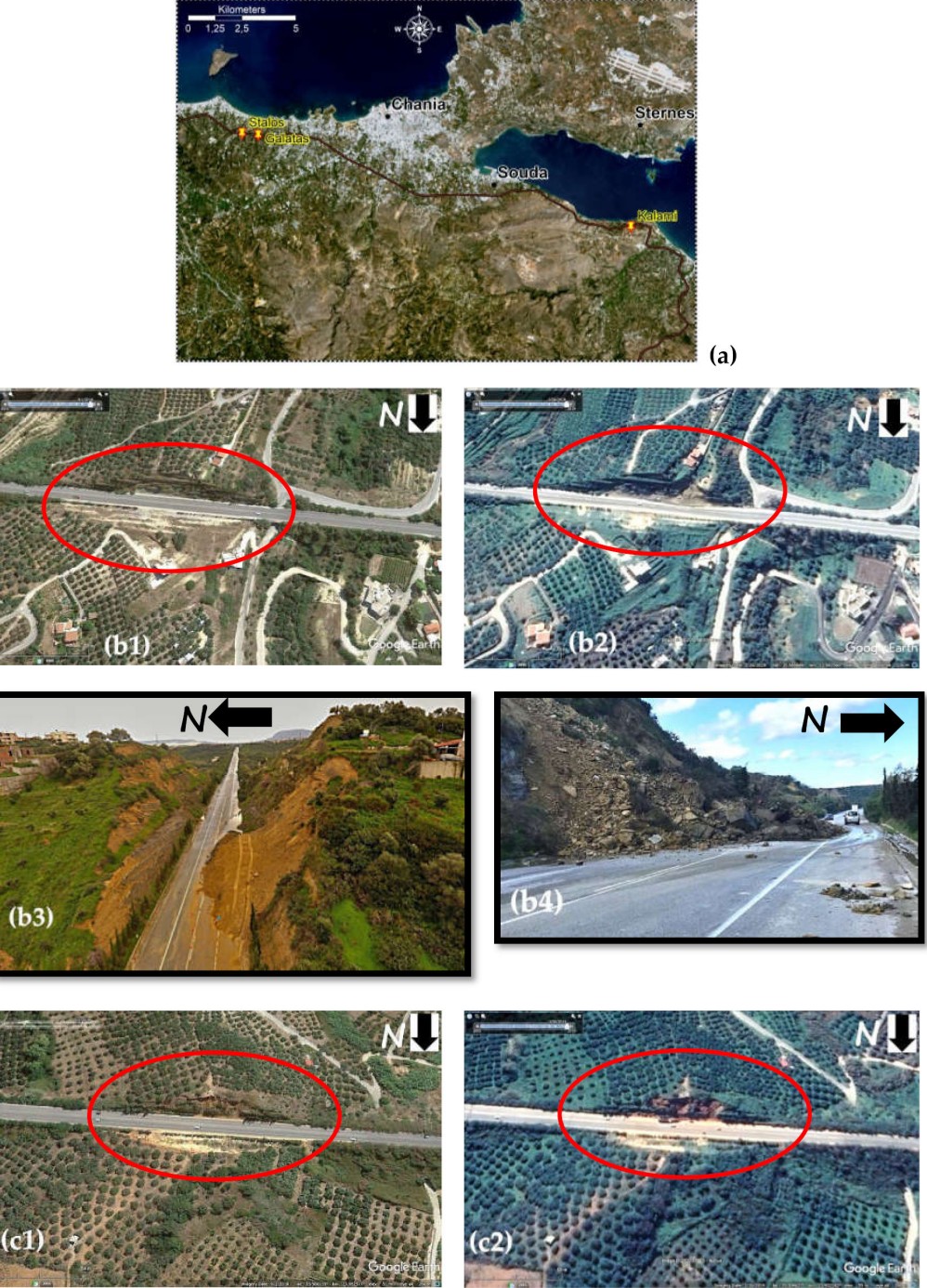

**Figure 4.** *Cont.*

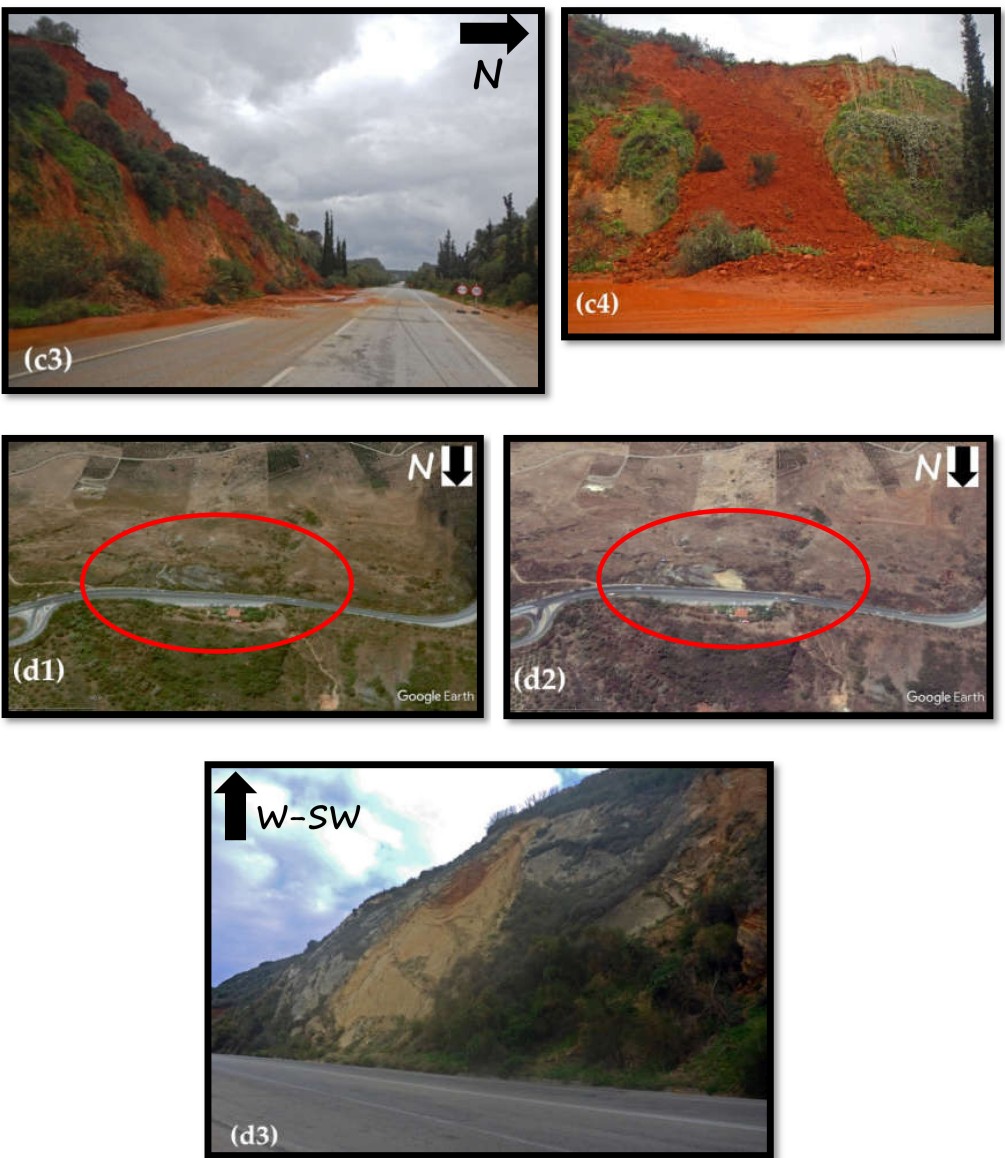

**Figure 4.** (**a**) The spatial distribution of three landslide events occurred recently along the motorway A90 due mainly to intense rainfalls, (**b1–b2**) Google Earth images showing the pre and post landslide event of Stalos area, (**b3–b4**) field survey photos of Stalos landslide event, (**c1–c2**) Google Earth images showing the pre and post landslide event of Galatas area, (**c3–c4**) field survey photos of Galatas landslide event, (**d1–d2**) Google Earth images showing the pre and post landslide event of Kalami area, (**d3**) field survey photos of Kalami landslide event.

### 3.2.2. Satellite Images Processing

Several EO data sets, free of charge, were acquired for this study. Sentinel-2 constellation of two satellites (A and B) constitutes the flagship of the European Space Agency (ESA) Copernicus program, which provides advanced and continuous data through the free access hub. Three S2 atmospherically corrected (2A) images, from the same season of the year (pre- and post-landslide, January 2018 and January-March 2019) were acquired. Even though the spatial analysis of Sentinel-2 images is considerably coarser than sub-meter detailed optical data, such as the VHR satellite images of WorldView, QuickBird, and so on, it is characterized by a much shorter revisit time and cost-free access [85,86]. The images were resampled to 10m (preprocessing procedure using as reference product band 2) and several false color composites (e.g., using Bands 3, 4, 8, 8A, 11 and 12) were tested to

distinguish the recent landslide scars. The Normalized Difference Vegetation Index (NDVI) was produced to assist the creation of the land cover map. NDVI denotes the variation among the visible red and the near infrared [87], relating to vegetation characteristics. The index shows values in the range−1 to +1. In the absence of vegetation (e.g., bare soil, rock, and urban areas), NDVI represents low positive (0 to 0.2) or negative values (all the other land covers except vegetation). On the other hand, in the existence of vegetation (e.g., grassland, forests), the index obtains values from 0.2 to 1 [88,89]. An image difference technique used to detect shallow landslide displacements between the pre- and post-landslide images. For this reason, two images from the same season were selected (January 2018 and January 2019) to avoid differences of vegetation phenology [90,91].

Moreover, two adjacent (path 182, row 35 and 36) Landsat 8 Level 2 images covering the study area, geometrically corrected, were obtained via the USGS Earth Explorer (EE) platform, for the detailed land cover map creation [92]. From these images, several products were created (NDVI, Normalized Difference Water Index, Tasseled Cap Transformation) to assist the creation of the land cover map. The processing of the data was made utilizing ENVI (5.5, Harris Geospatial Solutions, USA) and SNAP (7.0, ESA) software.

### 3.2.3. Digital Elevation Model

Terrain morphology is one of the critical parameters in the landslide phenomenon, and DEM constitutes a crucial information layer for various geomorphological products related directly to hazard evaluation [13,18,63,91]. With the base as the VHR aerial stereo imagery (1m), a sparse point cloud was generated initially for DEM extraction. This sparse point cloud was then used to create a dense point cloud. Point clouds generated from aerial scanners are better suited for detailed terrain analysis and feature extraction. Finally, an extremely accurate digital elevation model was built from this point cloud, with a pixel size of 2 m (Figure 5). The point cloud was generated using IMAGINE Auto DTM module of Hexagon Imagine Photogrammetry Suite, while the DEM was formed utilizing Global Mapper (20.0, Blue Marble Geographics).

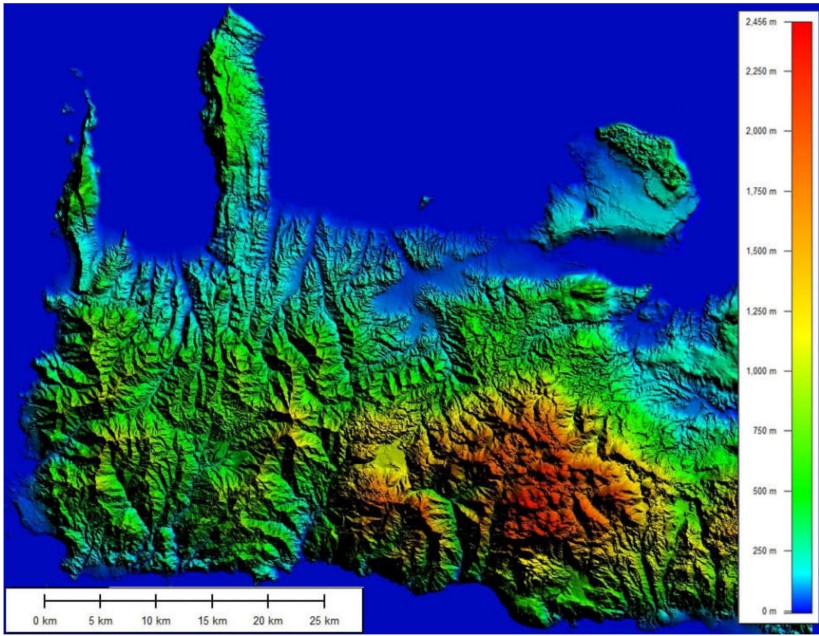

**Figure 5.** Very high-resolution Digital Elevation Model (DEM) (2 m) created from a dense point cloud derived from aerial stereo imagery.

### 3.2.4. Causal Factors—Geospatial Database

For the precise estimation of landslide-prone areas, the related causal factors need to be identified and validated, and a geospatial geodatabase should be built [27,93]. The choice of the critical factors in LS assessment usually is a fairly challenging issue, especially when detailed data are hard to be found. Hence, in the present research, eight evaluation factors comprising slope angle, geology, land cover, slope aspect, rainfall, distance to roads, distance to faults, and distance to streams were initially selected as evaluation factors [7,8,22]. Each factor was divided into sub-classes, and both factors and sub-classes were chosen according to a thorough literature review [8,22], extended field observations and scientific experience from previous studies. All layers were initially available in vector or raster formats and subsequently converted to a raster layout, having a spatial resolution of 2 m.

Topographic Factors

Topographic factors constitute the most significant geomorphological parameters that control slope stability [40]. From the VHR DEM, geomorphological thematic layers such as slope angle and aspect were calculated [40,94,95].

**Slope angle**

Slope angle and slope geometry are crucial issues for slope instability [17,40,43]. The moisture content, the pore pressure and the hydraulic behavior of a slope could be dominated by slope angle patterns [33,34]. Therefore, slope angle estimation is an essential part of hazard assessment. Slope angle was obtained from DEM and reclassified into five classes (based on our expert knowledge of the study area and related literature) according to their response to LS (a) very gentle slopes (0–10%) as 1, (b) gentle slopes (10–20%) as 2, (c) moderate steep slopes (20–30%) as 3, (d) steep slopes (30–60%) as 4, and (e) very steep slopes (>60%) as 5 (Figure 6a).

**Slope aspect**

Slope aspect defines the direction of slopes and reveals potential effects of prevailing winds, various weather conditions and incident solar radiation [25]. The orientations that according the local conditions receive more quantity or intense rainfall, the soil gets saturated more quickly, depending moreover on infiltration capacity, controlled by slope angle, soil type (permeability and porosity) and vegetation cover [80,94]. Slope aspect was grouped and reclassified into four classes, as (a) Flat areas and S-SE as 1 (b) NE-E as 2, (c) SW as 3 and (d) W-NW-N as 4 (Figure 6b).

Geological Units and Land Cover Factors

**Geology**

Geological characteristics are one of the most crucial controlling factors of landslides, since each geological unit have different susceptibility rates. Numerous researchers [2,56,96,97] emphasized the role of geology on slope stability. Geological formations and their relation to landslide phenomena directly based on their geomechanical characteristics, structural joints and discontinuities, hydrogeological behaviour, and indirectly the control of soil depth, vegetation cover erosion processes.

The geological layer extracted by digitizing the geological maps. The bedrock formations were reclassified into five general categories concerning LS (a) limestones and marbles as 1, (b) Neogene sediments as 2, (c) Loose Quaternary deposits as 3, (d) Phyllites–Quartzites as 4 and (e) flysch formations as 5 (Figure 6c).

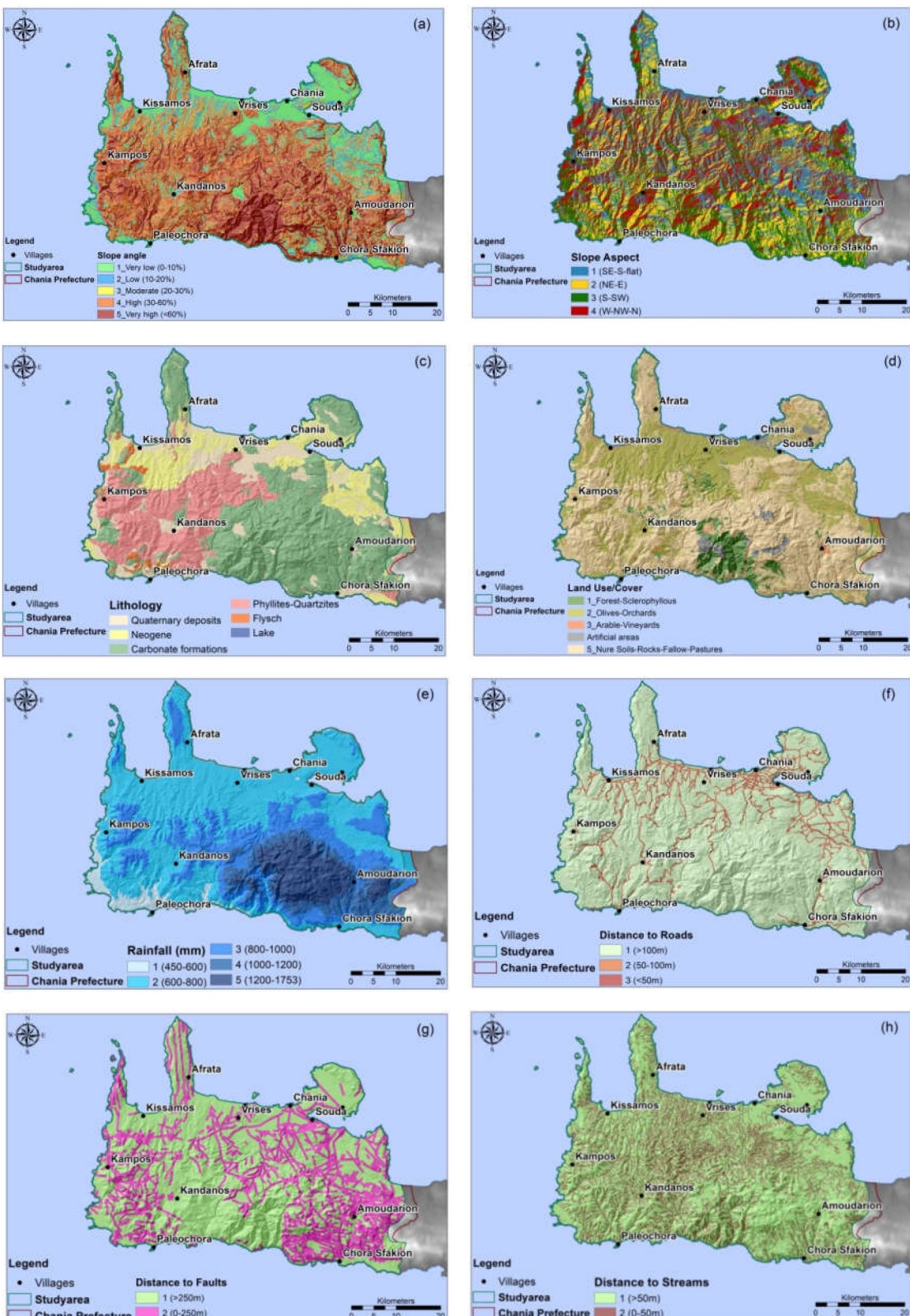

**Figure 6.** The thematic raster maps of the eight (8) factors, used for the estimation of Landslide Susceptibility of Chania Prefecture: (**a**) Slope angle, (**b**) Slope aspect, (**c**) Geology, (**d**) Land cover, (**e**) Mean annual precipitation distribution, (**f**) Distance to roads, (**g**) Distance to faults, (**h**) Distance to streams.

**Land Cover**

Land cover type is a potential driver of LS, as is vegetation, e.g., forest trees with a strong and extensive rooting system, and enhanced slope stability through their effect on the soil's hydrological and mechanical attributes [7,98]. The land cover map derived from the digitization of aerial orthophotos with the support of 15-m decorrelated images of Landsat 8 and the NDVI [NDVI = (NIR-R)/NIR+R)], where NIR-Near Infrared band 5 and R=Red band 4)] to improve the classification of vegetation cover. Finally, the cover was reclassified into five classes based on their response to LS (a) coniferous, broadleaved forest and shrubs as 1, (b) olive groves and orchards as 2, (c) arable land as 3, (d) urban areas as 4, (e) vineyards and bare soils as 5 (Figure 6d).

Mean Annual Precipitation

Rainfall distribution affects overland flow volume and soil moisture. In the present study the recommended World Meteorological Organization's (WMO) reference climatic period was used. The current climate reference period in use by WMO consists of 30 years from 1 January 1961 to 31 December 1990. However, it is also suggested that it is possible to adapt the reference period to following a "rolling" set of 30 years, updated every 10 years depending on best available datasets. Based on the above and considering the best available meteorological data series in the meteorological stations of the study area, in this application, the climatic reference period consists of 30 years from 1 January 1971 to 31 December 2000 [99,100]. Precipitation was spatially distributed based on its mean annual values, utilizing the co-kriging interpolation method and using a covariate as the basin elevation, as described in detail by Soulis et al. [79]. Mean annual precipitation was reclassified based on the rainfall amount into five classes (a) 0–600 mm as 1, (b) 600–800 mm as 2, (c) 800–1000 mm as 3, (d) 1000–1200 mm as 4, and (e) >1200 mm as 5 (Figure 6e).

Proximity Parameters

**Distance to roads**

Road constructions are related to extensive excavations, vegetation removal and, sometimes, the formation of steep slopes [24]. The main road network, consisting of primary (motorway and main urban streets), secondary (main provincial) and local (rest provincial) roads, was included as a potential triggering component and cause of landslide susceptibility. A buffer zone of 50 and 100 m over the main road network of the study area was delimited, and the distances to roads reclassified into three classes (a) >100 m as 1, (b) 50–100 m as 2 and (c) <50 m as 3 (Figure 6f).

**Distance to faults**

Proximity to tectonic structures generates the possibility of a landslide phenomenon since erosion processes and water flow along a crack might occur. For the faults derived from geological maps, a buffer zone of 250 m width was developed, and the area was divided into two sections, i.e., inside (as 2) and outside the zone (as 1) (Figure 6g).

**Distance to streams**

Vicinity to streams is an essential controlling factor of landslides since it can cause significant erosion processes (gully erosion) [2,101]. Second or higher order streams according to Strahler's (1964) [102] classification were selected and processed by creating a 50 m buffer zone. Thus, the basin area was divided into two sections, i.e., within (as 1) and outside (as 2) the zone (Figure 6h).

3.2.5. Weights Factor Analysis based on AHP

A semiquantitative multi-criteria method, AHP, in which weights are used to make decisions via pair-wise relative evaluations with no inconsistencies, was used in this procedure. The technique has been effectively utilized in LS mapping by several researchers [39,40,43,44]. AHP includes a hierarchical process of decision factors and creates comparisons between possible pairs in a matrix to provide a weight for each factor and correspondingly a consistency ratio [103].

Thus, the significance of each factor comparative to every single other factor is described ideally. In AHP all factors are rated utilizing a continuous scale from 1 to 9, when there is a direct correlation between the factors (1: Equal importance; 3: Moderate prevalence of one over another; 5: Strong or essential prevalence; 7: Very strong or demonstrated prevalence; 9: Extremely high prevalence; 2, 4, 6, 8: Intermediate values; Reciprocals: opposites, for inverse comparison, used to represent compromises between the preferences in weights 1, 3, 5, 7 and 9) if the factors have direct relation (when the factor on the vertical axis is more important than the factor on the horizontal one) and a scale from 1/2 to 1/9, when there is an inverse correlation between the factors (Table 2) [45,104].

**Table 2.** The landslide causative factors' pair-wise comparison matrix and weights (normalized principal eigenvector) for based on Analytical Hierarchy Process (AHP) method implementation.

| | (1) | (2) | (3) | (4) | (5) | (6) | (7) | (8) | Weights (*100) |
|---|---|---|---|---|---|---|---|---|---|
| Slope angle (1) | 1 | 3 | 3 | 5 | 5 | 7 | 7 | 7 | 17.7 |
| Geology (2) | 1/3 | 1 | 1 | 2 | 5 | 7 | 7 | 9 | 6.0 |
| Mean annual precipitation (3) | 1/3 | 1 | 1 | 3 | 3 | 5 | 8 | 7 | 7.3 |
| Land Cover (4) | 1/5 | 1/2 | 1/3 | 1 | 3 | 5 | 9 | 7 | 5.4 |
| Distance to Roads (5) | 1/5 | 1/5 | 1/3 | 1/3 | 1 | 2 | 5 | 5 | 2.6 |
| Slope Aspect (6) | 1/7 | 1/7 | 1/5 | 1/5 | 1/2 | 1 | 2 | 2 | 0.8 |
| Distance to faults (7) | 1/7 | 1/7 | 1/8 | 1/9 | 1/5 | 1/2 | 1 | 2 | 1.2 |
| Distance to streams (8) | 1/7 | 1/9 | 1/7 | 1/7 | 1/5 | 1/2 | 1/2 | 1 | 1.1 |
| | | | | | CR = 0,064 | | | | |

An essential feature of the AHP is that it allows us to define rating inconsistencies by the consistency index (CI), which is used as specified by Equation (1) [44,105]:

$$CI = \frac{\lambda_{max} - N}{N - 1} \tag{1}$$

where $\lambda_{max}$ is the biggest eigenvalue and N is the order of comparison matrix.

Saaty (1980) [44] established the consistency ratio (CR) which is determined as the ratio among the consistency index (CI) and the average random consistency index (RI) for different matrix orders [CR = CI/RI]. The CR depends on the number of factors. In the case that CR is higher than 0.1, the comparison matrix is inconsistent and should be revised. In the present study, the CR value is 0.064 (< 0.1), which makes definite the great consistency between the preferences that are applied to create the comparison matrices (Table 2) [39,40,45].

3.2.6. Landslide Susceptibility Index (LSI)

The different rating values were assigned to the classes of the thematic layers as attribute records in a GIS environment, and a raster map was developed for each final weighted layer. The maps were reclassified to be compared utilizing the equal interval method, in grades from very low to very high susceptibility. Subsequently, the reclassified raster layers were applied as input data for the LSI estimation. The weights were multiplied by 100 to achieve integer numbers and obtain values between 0 and 100. Subsequently, LSI was calculated by multiplying the assigned level values of the raster layers (of the 8 factors) with the corresponding weights derived from the AHP method.

Two different calculations were made, one for the estimation of LSI for all the study area (Chania Prefecture), by multiplying the thematic layer grids with the corresponding weights (wt1 in Table 3, Equation (2)), and another for the estimation of LSI(m) along the A90 motorway giving higher importance to slope angle and multiplying by the corresponding slope angle-based weights (wt2 in Table 3, Equation (2)).

$$LSI \text{ and } LSIm = \sum_{i=1}^{n} weight_i \text{ x class rate} \tag{2}$$

where n is the total number of data layers (factors) and i is the different weight (wt1 and wt2 for the entire area and A90 correspondingly) in Table 3.

**Table 3.** Classification of the landslide controlling factors and categorization into rating classes according to their significance on Landslide Susceptibility.

| Factor | Class | Class Value Rating | Weight wt1 | Weight wt2 |
|---|---|---|---|---|
| **i. Slope Angle (SA) (%)** | 0–10 | 1 | 17.7 | 17,7*SA*(MAP*2,0) layer |
| | 10–20 | 2 | | |
| | 20–30 | 3 | | |
| | 30–60 | 4 | | |
| | >60 | 5 | | |
| **ii. Aspect** | Flat, S-SE | 1 | 0.8 | 0.8*SA*(MAP*2,0) layer |
| | E-NE | 2 | | |
| | SW | 3 | | |
| | W-NW-N | 4 | | |
| **iii. Geology** | Limestone–marble | 1 | 6.0 | 6.0*SA*(MAP*2,0) layer |
| | Neogene | 2 | | |
| | Loose quaternary deposits | 3 | | |
| | Phyllites–Quartzites | 4 | | |
| | Flysch | 5 | | |
| **iv. Land Cover** | Forest and shrubs | 1 | 5.4 | 5.4*SA *(MAP*2,0) layer |
| | Olive groves–Orchards | 2 | | |
| | Arable crops | 3 | | |
| | Urban areas | 4 | | |
| | Nude soils and vineyards | 5 | | |
| **v. Mean annual precipitation (MAP) (mm)** | 0–600 | 1 | 7.3 | 7.3*SA *(MAP*2,0) layer |
| | 600–800 | 2 | | |
| | 800–1000 | 3 | | |
| | 1000–1200 | 4 | | |
| | >1200 | 5 | | |
| **vi. Distance to Roads (m)** | >100 | 1 | 2.6 | 2.6*SA *(MAP*2,0) layer |
| | 50–100 | 2 | | |
| | <50 | 3 | | |
| **vii. Distance to faults (m)** | >250 | 1 | 1.2 | 1.2*SA*(MAP*2,0) layer |
| | <250 | 2 | | |
| **viii. Distance to streams** | >50 | 1 | 1.1 | 1.1*SA *(MAP*2,0) layer |
| | <50 | 2 | | |

## 4. Results

### 4.1. Landslide Susceptibility Index—Study Area (Chania Prefecture)

For the eight selected factors related to landslide events in the area under study, a statistical assessment was made to investigate the distribution and relation of past landslides to the sub-classes of each factor. From the analysis of the past landslide distribution, it was obtained that: (a) in the slope angle map, the 38% and 47.5% of previous LS are situated in very high and high slope angles, respectively; (b) in the geological map, 63.3% of past LS parts are located in Phyllites–Quartzites and 21.1% in limestones; (c) in the land cover map, 65.2% of previous LS are placed in bare soils, while 30.2% and 10.1% in olive groves and forest–shrub areas, respectively; (d) in the mean annual precipitation distribution map, 77.7% of past LS parts are situated in 600–800 m, and 800–100 mm classes; (e) in the aspect map 31% and 30% of past LS is located in N-NW-W, and NE-E directions; (f) in the distance to roads, streams and faults maps, 57.8%, 61% and 70% of the past landside events are placed in areas >100 m, >50 m and >250 m classes, respectively.

The LSI map is classified into five susceptibility classes, defined as (a) Very Low, (b) Low, (c) Moderate, (d) High and (e) Very High. Higher values of LSI demonstrate more susceptible areas to landslides. The generated map was processed (dissolved and smoothed out using a 5x5 low pass filter) to decrease the significance of possible misclassified cells. Approximately 15.30% of the prefecture shows very high LSI values, 30.25% show high values, 36.94% show moderate values, while 15.50% and 2.02% appear to have low and very low values, respectively (Figures 7 and 8a).

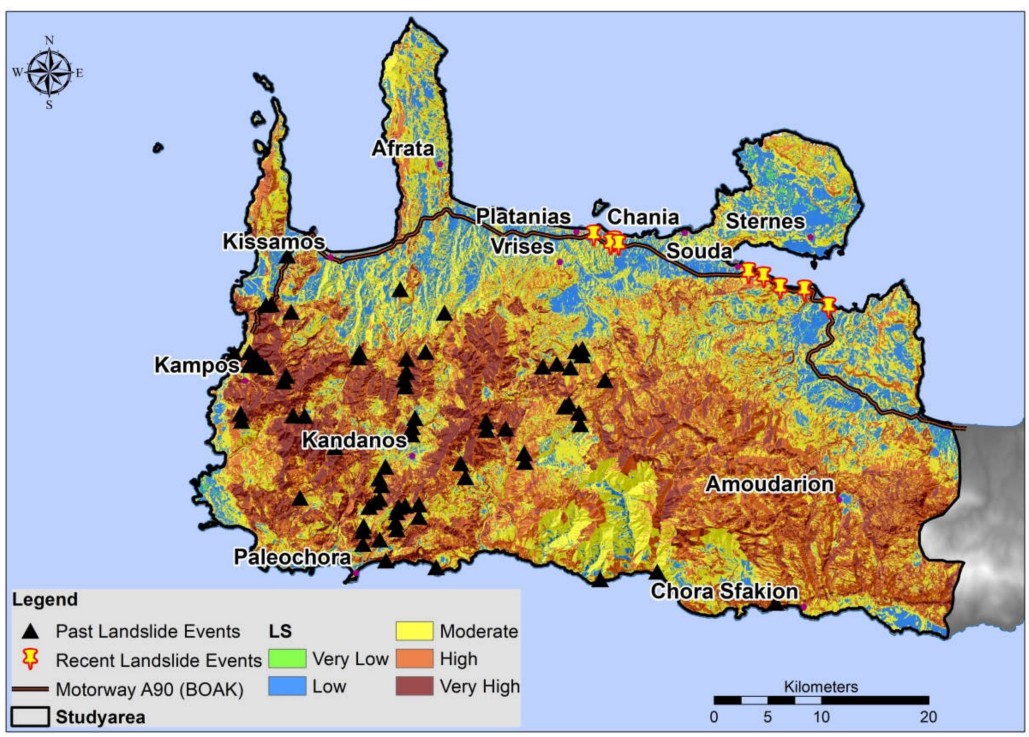

**Figure 7.** The Landslide Susceptibility (LS) map, after reclassifying the calculated values into five classes of potential susceptibility.

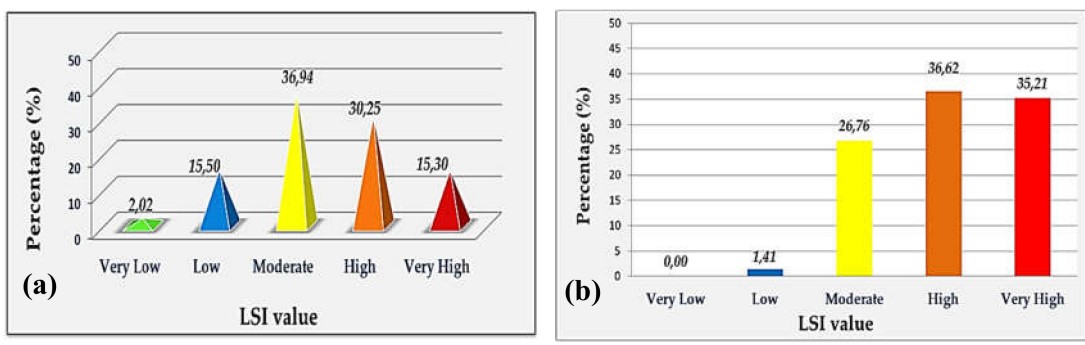

**Figure 8.** (**a**) The percentage of the five LSI classes in the area under study, (**b**) the percentage of the past landslides in correlation with the LSI map.

Additionally, from the GIS-based statistical analysis, it was found that the most susceptible areas to landside (high and very high values) are placed to (a) slope angles with very high and high values (42.17%), (b) Geological formations of Phyllites–Quartzites (38%) and limestone–marble (49%), (c) land covers of bare soils (87%), (d) Slope aspects of W-NW-N (34%) and SW (28%), (e) Mean annual precipitation >800 mm (60%), (f) Distance from roads >100 m (79%), (g) Distance to faults >250 m (55%), (h) Distance to streams >50 m.

The LS calculations are geospatial analytical types and should be assessed compared to the info utilized to formulate the estimation. The final stage comprises the validation procedure. To make the comparison, the separation of the past landslide data in the area under examination is needed. Once the data are partitioned, one subset of data is used to obtain the prediction image while the other subset is compared with the prediction results for validation. Thus, the validation was achieved by comparing the generated LSI maps with the selected landslide data that were initially set aside for validation (20%) from the total geodatabase and landslide-free chosen areas in the region (another 16 points—in total, 32). The GIS-based comparison and statistical analysis showed adequate results since the overall matching of the five classes reach a percentage of 83.7% (Table 4). The past reference landslides are distributed with a percentage of 25% in moderate and 69% in high and very high susceptible areas (Figure 8b).

**Table 4.** Confusion matrix of LS map validation.

| Validation Sample | Target Class (Observed) | |
| --- | --- | --- |
| | Susceptible Areas (High and Very High Classes) | No Susceptible Areas (Moderate, Low, Very Low Classes) |
| Landslide areas | 11 | 5 |
| Landslide-free areas | 2 | 14 |

### 4.2. Landslide Susceptibility Index(LSIm)—A90 Motorway

As previously mentioned, the LSIm map was created only for the northern part of the prefecture, along the motorway A90, by giving higher importance to slope angle (due to the steep slopes across the main road network) and mean annual precipitation (multiplied by 2.0 in the calculation, due to the recent intense and high amounts of rainfalls) factors which, according to the thorough knowledge of the area and the extended field visits, appear as the two primary triggering parameters of the recent landslide events. In consequence, the final extracted map was also classified into five susceptibility categories, (a) Very Low, (b) Low, (c) Moderate, (d) High and (e) Very High. The GIS-based comparison and statistical analysis showed an overall matching of the five classes that reaches a percentage of 89.2% The statistical analysis of the recent events occurred alongside the A90, reveals that 56.6% of the events took place in the high and very high and 44.4% in the moderate susceptible areas of LSI map.

With the goal to explore the potential landslide hazard associated with other susceptible points of motorway A90, a thorough analysis of LSIm and Slope angle map was accomplished. From this analysis, several other regions were chosen, and four of them delineated in Figure 9. Two of these (1–2) are related to the recent landslides that occurred in 2018–2019 period, while the other two (3–4) reveal some spots (yellow arrows in Figure 9) with high and very high landslide susceptibility and potential hazard. More specifically, these spots appear to have slope angle values higher than 60% and very high susceptibility values are due principally to their geological and geomorphological characteristics and additionally to the high amount of rainfall during the previous mentioned period (2018–2019).

### 4.3. EO Data in Landslide Detection

In the present research, a comprehensive analysis of EO data was made to detect the recent landslide scars along the A90 motorway at the north part of Chania prefecture. Since the two images were acquired from the same season of the wear (January 2018 and 2019), the vegetation phenology remains the same.

To display slope movements, several RGB composites were made, but the most sufficient was the combination of near infrared Band 8 in red and the shortwave infrared Bands 12 and 11 in Green and Blue colors, respectively (Figure 10). This RGB composite image revealed small landslide scars in the Stalos and Galatas regions (Figure 10–cyan polygon). Landslide scars were distinctly different from the surrounding ground features and appear in dark colors, since the surrounding area appears in red (vegetation cover). The detected landslide areas in the Stalos and Galatas regions cover an area of

approximately 1474 m² and 1020 m², respectively. On the other hand, from the analysis of the NDVI difference image (NDVI post—NDVI pre-event, created by Sentinel-2 images), the landslide scars were difficult to distinguish, probably because the change in vegetation cover was not so intense.

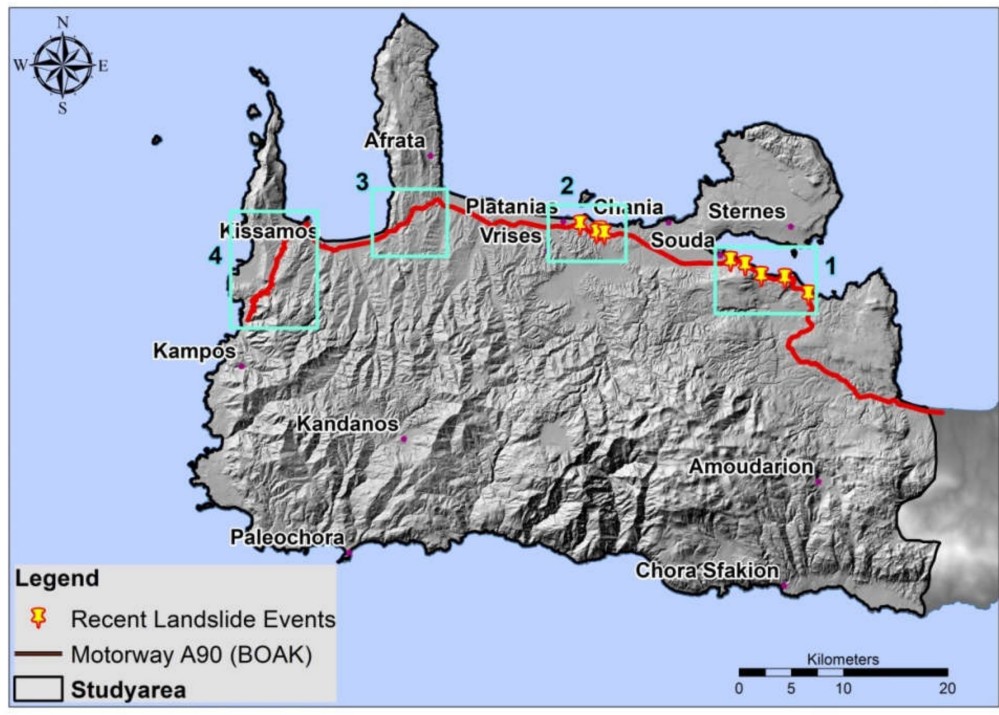

(**a**)

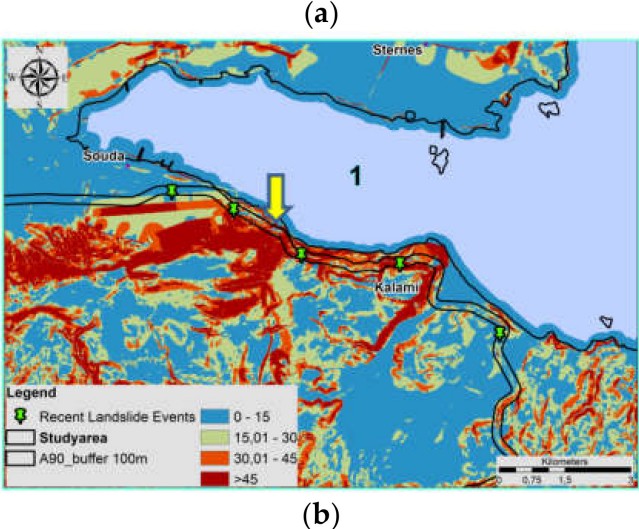

(**b**)

**Figure 9.** *Cont.*

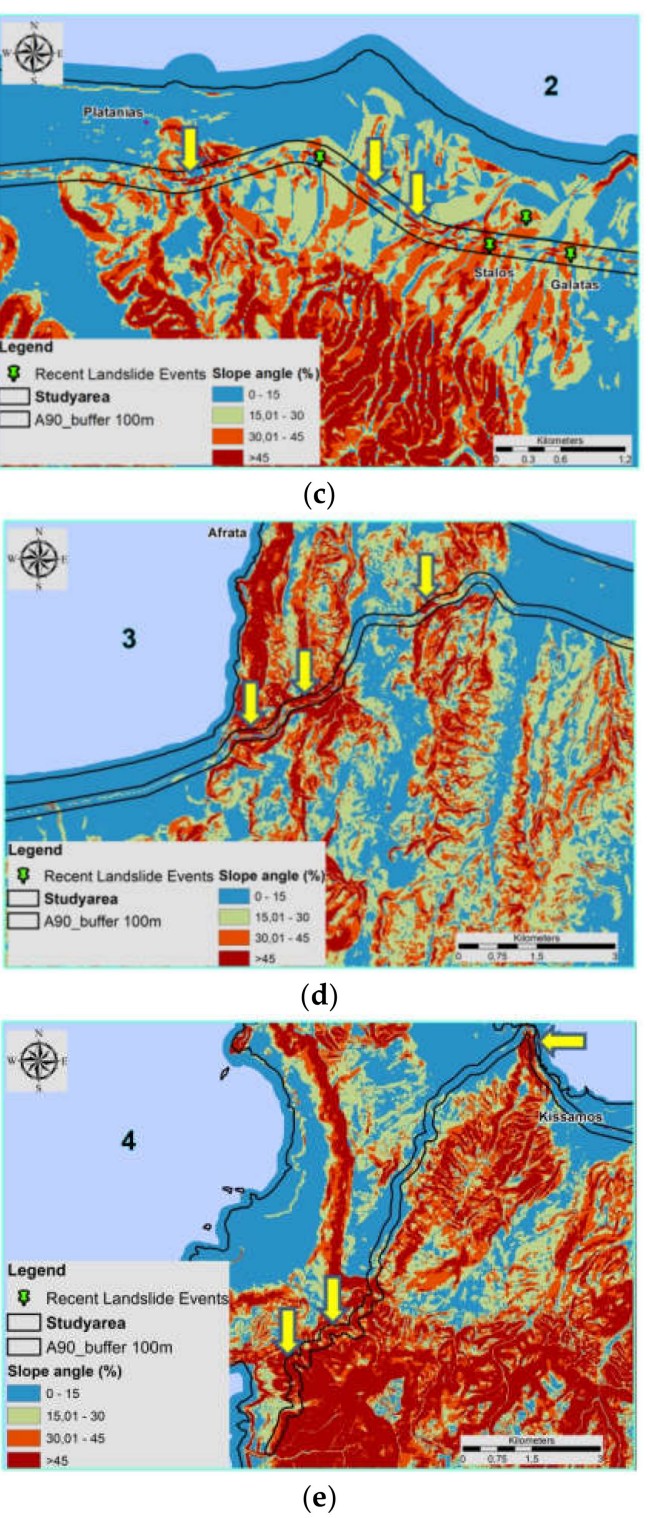

**Figure 9.** (**a**) Perspective view of the four selected regions distribution, (**b**–**c**) the selected regions 1-2 where the recent landslide events occurred, (**d**–**e**) the selected regions 3-4 with very high landslide susceptibility and potential hazard (critical spots indicated with yellow arrows).

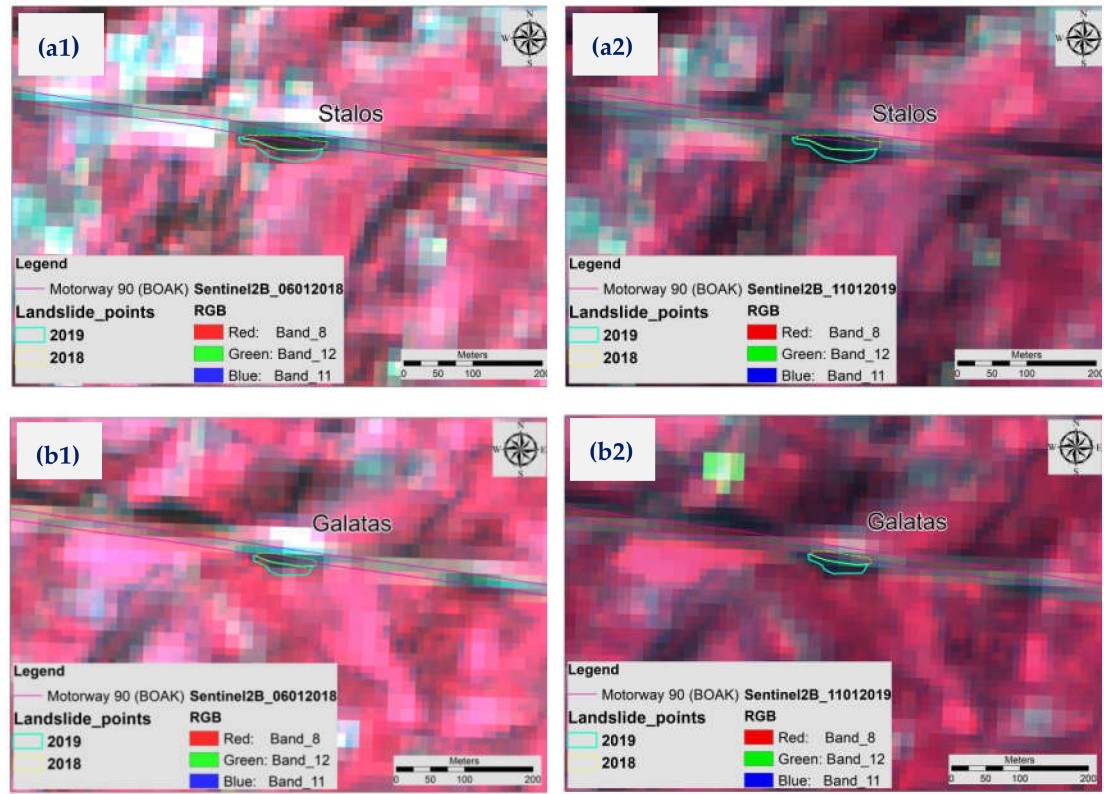

**Figure 10.** False color composites of Sentinel-2 pre (**a1,b1**) and post-landslide (**a2,b2**) events in Stalos and Galatas areas of motorway A90. Cyan polygons represent the landslide scars.

## 5. Discussion

Landslides are recognized as one of the most disastrous natural hazards, and are affected by several triggering parameters which differ significantly from place to place. Several factors, such as geological formations, geomorphology, rainfall and human activities, are capable of changing landscape features [2,5] decisively. This goal of this study is to deal with the analysis of the various physical parameters, such as geological formations, geomorphology and rainfall, as well as anthropogenic factors (primarily road constructions and their characteristics, e.g., steep slopes) on LS at the Chania prefecture. LS assessment is an essential task in the landslide management process, and their estimation should be performed formerly of all kind of manufacturing projects and constructions.

Two different kinds of LS maps regarding Chania prefecture and the entire length of the motorway (A90) inside the area under investigation were created. To that end, eight influencing factors were chosen (slope angle, slope aspect, geology, land cover, mean annual precipitation distribution, distance from roads, distance from faults and distance from streams). From the available methods, the semi-quantitative approach (AHP-WLC) was selected, which is easy to apply and accurate for LS mapping. The pair-wise comparisons of the relative factors are calculated without discrepancies in the decision process. Moreover, the rating values in this study may be used in other study areas with similar geological, geomorphological and hydro-climatic conditions. The AHP method was applied to obtain a factor's weight values, and then by using the WLC process, the LSI was estimated for every pixel.

The results reveal the distribution and characteristics of the phenomena in the study area accurately. The LSI map (Figure 7) shows that 45% of the prefecture belongs to high and very high susceptibility conditions and are situated mainly at the south and southwestern part. The landslides are related mostly to the landslide-controlling parameters of geology (Phyllites–Quartzites and limestone–marbles), natural geomorphological characteristics and less with the mean annual precipitation distribution,

aspect, land cover, etc. The types of the landslides, in this case, are translational, rotational and rockfalls [22,23]. On the contrary, the LSIm map results depict that almost 70% of the prefecture's northern part, adjacent to the main road network (motorway A90), belongs to high and very high susceptibility areas. The triggering parameters of the recent landslides in this region, which was almost a landslide-free area in the past, are associated with the continuous and intense rainfalls (from the September 2018 to February 2019 period) [75], and the anthropogenic activity and, more specifically, the steep slopes along the constructed road network (this slope angle and distance to road parameters). For this reason, most of the landslide movement types are related to the debris avalanche type. To verify the accuracy and suitability of LSI maps, the landslide inventory map was created based on accessible landslide historical data, remote sensing images (aerial orthophotos and satellite images), and fieldwork records. The statistical analysis and validation process proved that the produced LSI map was satisfactorily accurate and adequate to detect landslide-prone areas in other regions along the motorway A90, and therefore is characterized as a significant basis for the estimation of landslide hazard.

EO data were utilized for land cover mapping and landslide scar detection. The Sentinel-2 image processing results in landslide detection are determined by a few other issues, which regard the landslide size (scar), and the ground conditions. Considering that the spatial resolution of Sentinel-2 is limited to 10 m, i.e., landslide slip surface smaller than the pixel size (10mx10m), it is not easily detectable. On the other hand, indeed, this technique and resolution are adequate to distinguish sizable landslides. Moreover, as far as it concerns the vegetation cover of the area that changed by slope movement, if the alteration is not so drastic or vegetation recovery is quicker than the observation cycle (which also depends on cloud cover), it would be problematic for the vegetation difference technique to detect it. Hence, only two of the recent landslides were mapped. Nevertheless, it is certain, that Sentinel-2 high resolution, five-day revisit and cost-free dataset should be indisputably utilized from the scientific community in future landslide detection research, especially in more extensive events [85,86,91].

Regarding the effort to provide knowledge for the prevention and mitigation of new catastrophic landslides in the future, and according to results of this study, we suggest that in areas adjacent to already constructed and future infrastructure projects, where the potential landslide vulnerability is very high, appropriate protection measures should be taken to avoid fatalities of human life and serious economic and social impacts. These actions may be active or passive, depending on the purpose they serve and the scale of the phenomenon to be remedied. Therefore, in areas where landslides have occurred or could potentially occur, the following remedial measures are suggested and indicative by event factor: (a) Slopes (i) proper slope shaping, (ii) active and passive protection measures such as anchorages (passive or active) grids and restraint fences, respectively, (iii) protection against corrosive phenomena and conservation measures for drainage of surface and groundwater, (iv) construction of small-scale construction projects (retaining walls, rock traps) but also larger-scale construction (mounting brackets, defense arcades, etc.); (b) Geological parameters dealing with the implementation of the appropriate geological and geotechnical research to identify and delimit potentially unstable areas; (c) Unforeseen Factors related to (i) weather conditions (e.g., increased rainfall amounts and intensity), (ii) human activities and (iii) natural disasters (e.g., fires have an impact on the occurrence of landslides).

## 6. Conclusions

The objective of the present study was the implementation of the semi-quantitative method of AHP-WLC, to create a detailed LS map for (i) Chania prefecture and (ii) the northern part of the county along the main road network motorway A90, so as to reveal the predominant landslide-triggering factors, the susceptibility and the potential future hazard of the area to similar phenomena. A GIS-based statistical analysis, along with the synergistic use of EO data and detailed DEM created from a dense

point cloud, was applied to determine the relation of physical and anthropogenic parameters with landslide activity in the region.

Very high-resolution aerial photos, satellite imagery (Google Earth in our case) and DEM provide significantly powerful tools for landslide detection and extraction of necessary detailed layers such as slope angle and land cover. Despite the coarse spatial analysis of Sentinel-2 images in comparison with other very high-resolution satellites, it provides a powerful tool for landslide detection and monitoring because of its high temporal resolution (five days revisit time) and free availability.

The LSI map results reveal the vast extent of susceptible areas in Chania prefecture as well as along the highway A90, where several potential hazard points were assessed. Moreover, it depicts the different characteristics of past and recent landslide events, with different preparatory and triggering mechanisms, where the former depends on geological and geomorphological factors and the latter on extreme rainfall conditions and anthropogenic activities.

The LSI map findings could be useful for managers and decision-makers in landscape administration and land use design in the area under investigation. Particularly in the case of the motorway A90, further studies should be performed at the local level to clarify the accurate extent site of the slope vulnerability and landslide hazard. In general, it is confirmed that the synergistic use of earth observation and geospatial analysis systems can ideally assist research regarding landslide susceptible areas.

**Author Contributions:** Conceptualization, E.P. and A.P.; methodology, E.P. and A.P.; software, E.P. and K.X.S.; validation, E.P., A.P., K.X.S. and I.C.; formal analysis, E.P., A.P. and K.X.S.; resources, E.P., A.P., K.X.S., I.C. and D.-S.A.; writing—original draft preparation, E.P., A.P., K.X.S. and D.-S.A.; supervision, E.P.; field visits, A.P. All authors have read and agreed to the published version of the manuscript.

**Funding:** This research received no external funding

**Conflicts of Interest:** The authors declare no conflict of interest.

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
