# Peer review of "Landslide Mapping and Susceptibility Assessment Using Geospatial Analysis and Earth Observation Data"

_land, doi:10.3390/land9050133_

Round 1

Reviewer 1 Report

The article under review is the result of an interesting study, as it deals with a very catastrophic phenomenon of soil failure due to natural processes and anthropogenic interventions, which is of interest not only to the study area but any area of ​​land prone to the phenomenon.
The overall view that a reviewer takes from the structure of the article is that it follows a strictly scientific approach to methodology, has all the distinctive structural elements that an article should satisfy, refers to a fairly rich bibliography and a documented presentation of data used and not only in conclusions that explain the correctness of the method, but also guidelines that must be taken to address the phenomenon of landslides in a timely manner.
For all of the abovementioned reasons, my point of view is that this article can be published without corrections.

Best regards,

Author Response

Thank you very much for your comments

Reviewer 2 Report

Dear Authors,

I read your paper about LSM By GIS techniques in the Chania prefecture of Crete Island.

The paper is clear and quite well written, I have only few remarks about it.

the first one is about the novelty, which is quite low, since several scientific papers on the same topic exist, even with more advanced methodologies to define LSM.

It is not clear how you defined the number of classes of the parameters you use and why you used different number of classes of each parameter.

The result of AHP are not clear, it looks like that distance to faults and to streams are more important that rainfall, but at the beginning of the paper you described several rainfall induced landslides.

Concusion and Discussions sections should be divided.

Please find detailed comments in the attached document.

Best Regards 

Reviewer 3 Report

Dear Authors, I have read your manuscript "Landslide mapping and risk assessment using geospatial analysis and earth observation data". I think there are several issues that need to be addressed. Hopefully, after major revisions the manuscipt could be maybe publishable. 

GENERAL COMMENTS

-Risk analysis is not encompassed in your study. Please note that a complete risk assessment encompasses the definition (and combination) of hazard (probability of occurrence), vulnerability (degree of expected loss) and exposure (worth of elements exposed). Your study investigates only the spatial component of hazard. It would be better to refer only rto "susceptibility mapping" activity. Moreover, please note that hazard and risk are not synonimous. Please remove all references to risk in your manuscript or substitute the term "risk" with "hazard", or "susceptibility".

-By reading the title of some cited articles, sometimes the references do not seem very appropriate.

-Why the mean annual precipitiation definition is limited only to 1971-2004? More than 15 years of records are missing and I think that given the recent trends of climate change the missing data may be relevant.

-It is important that you clearly define which is the landslide typology considered in your database and thus in your study. Very different landslide typologies have different predisposing factors, thus they cannot be analyzed together. 

-The hazard/susceptibility maps have not been quantitatively validated. Despite the claims of the conlusions, I don't see ROC analyses, confusion matrixes, skill scores or other quantitative indexes commonly used to validate susceptibility maps. This is a critical flaw that needs to be fixed. 

-A discussion of the results is completely missing. Chapter 4 is only a recap of the work and a list of very generic remarks, thus resembling more a conclusion and not a discussion.

SPECIFIC COMMENTS

References 2-5 do not seem very appropriate to support such a general sentence. I suggets to make reference to something more recent and more general like: Froude, M. J., & Petley, D. (2018). Global fatal landslide occurrence from 2004 to 2016. Natural Hazards and Earth System Sciences18, 2161-2181.

L49. I suggest to add also: Mendes, R. M., de Andrade, M. R. M., Tomasella, J., de Moraes, M. A. E., & Scofield, G. B. (2018). Understanding shallow landslides in Campos do Jordão municipality-Brazil: disentangling the anthropic effects from natural causes in the disaster of 2000. Natural Hazards & Earth System Sciences18(1).

L51 I suggets to add something more recent like: Reichenbach, P., Rossi, M., Malamud, B. D., Mihir, M., & Guzzetti, F. (2018). A review of statistically-based landslide susceptibility models. Earth-Science Reviews180, 60-91.

Some of the four objectives (3 and 4) do not seem well centered, as only general comments are reported in the discussion and conclusion and are not supported by data.

L215 You listed six of them

L228 please avoid repetitions

Fig 4 I think there are too many panels in this figure. Please, consider selecting only the most relevant pictures.

L290 constitute

L311 About the importance of geology in hazard studies, I suggets to add also: Segoni, S., Pappafico, G., Luti, T., & Catani, F. (2020). Landslide susceptibility assessment in complex geological settings: sensitivity to geological information and insights on its parameterization. Landslides, 1-11.

L320 Why didn't you use Corine Land Cover dataset? It is free, robust and spatially defined over the whole Europe.

L329 I suggest (here and elsewhere) to name this factor "mean annual precipitation" instead of "rainfall", because the latter is too generic.

L450 are due

Section 3.3 I don't understand the rationale of this section. It seems quite obvious.

Round 2

Reviewer 2 Report

Dear Authors,

thank you for your replies to my comments.

I have only few remarks about your work.

Since the division in classes of each parameters is not useful for the LS mapping, i suggest to present your maps with a stretched colors visualization, at least for numerical variables, since the way you presented these maps can led to some misunderstanding from the reader. Furthermore, your response "The classes were set according to the knowledge of each factor characteristics in the study area and their relation to landslide susceptibility" is not acceptable, since your expertise about the study area is useful to define which parameters have to be considered, but to define the number of classes, several statistic rules exist.

Regards

Reviewer 3 Report

The manuscript has been improved and my comments have been addressed. 

I suggest minor revision to better check the text: some small errors are present and some sentences may be improved.   

Author Response

This manuscript is a resubmission of an earlier submission. The following is a list of the peer review reports and author responses from that submission.